# The effect of transdermal gender-affirming hormone therapy on markers of inflammation and hemostasis

**Moya H. Schutte**[1], **Robert Kleemann**[2,3], **Nienke M. Nota**[1], **Chantal M. Wiepjes**[1], **Jessica M. Snabel**[2], **Guy T'Sjoen**[4], **Abel Thijs**[5], **Martin den Heijer**[1] *

**1** Department of Endocrinology and Center of Expertise on Gender Dysphoria, Amsterdam University Medical Centers, Amsterdam, The Netherlands, **2** Department Metabolic Health Research, The Netherlands Organisation for Applied Scientific Research (TNO), Leiden, The Netherlands, **3** Department of Vascular Surgery, Leiden University Medical Center, Leiden, The Netherlands, **4** Center for Sexology and Gender, Ghent University Hospital, Ghent, Belgium, **5** Department of Internal Medicine, Amsterdam University Medical Centers, Amsterdam, The Netherlands

* m.denheijer@amsterdamumc.nl

**Data Availability Statement:** All relevant data are within the paper and its Supporting information files.

## Abstract

### Background

Cardiovascular risk is increased in transgender persons using gender-affirming hormone therapy. To gain insight into the mechanism by which sex hormones affect cardiovascular risk in transgender persons, we investigated the effect of hormone therapy on markers of inflammation and hemostasis.

### Methods

In this exploratory study, 48 trans women using estradiol patches plus cyproterone acetate (CPA) and 47 trans men using testosterone gel were included. They were between 18 and 50 years old and did not have a history of cardiovascular events. Measurements were performed before and after 3 and 12 months of hormone therapy.

### Results

After 12 months, in trans women, systemic and endothelial inflammatory markers decreased (hs-CRP -66%, (95% CI -76; -53), VCAM-1 –12%, (95% CI -16; -8)), while platelet activation markers increased (PF-4 +17%, (95% CI 4; 32), β-thromboglobulin +13%, (95% CI 2; 24)). The coagulation marker fibrinogen increased transiently, after 3 months (+15%, (95% CI 1; 32)). In trans men, hs-CRP increased (+71%, (95% CI 19; 145)); platelet activation and coagulation markers were not altered. In both trans women and trans men, leptin and adiponectin changed towards reference values of the experienced gender.

### Conclusions

Platelet activation and coagulation marker concentrations increased in trans women using transdermal estradiol plus CPA, but not in trans men using testosterone. Also,

**Funding:** The authors received no specific funding for this work.

**Competing interests:** The authors have declared that no competing interests exist.

concentrations of inflammatory markers decreased in trans women, while hs-CRP increased in trans men. Our results indicate that hormone therapy may affect hemostasis in transgender persons, which could be an underlying mechanism explaining the increased cardiovascular risk in this population.

## Introduction

Transgender persons experience an incongruence between their sex assigned at birth and their gender identity. This is opposed to cisgender persons, whose sex assigned at birth matches their gender identity. Transgender persons can receive gender-affirming hormone therapy (GAHT) as part of their transition. In trans women (male sex assigned at birth, female gender identity), hormone treatment consists of estrogens, often in combination with antiandrogens (in Europe usually cyproterone acetate, CPA). In trans men (female sex assigned at birth, male gender identity), hormone treatment consists of testosterone [1]. Previous studies have shown that both trans women and trans men receiving hormone therapy have an increased risk of cardiovascular events compared to the general population [2, 3]. Trans women have an increased risk of stroke, myocardial infarction, and venous thromboembolism. Trans men seem to have an increased risk of stroke and myocardial infarction [4, 5].

The pathophysiological mechanism by which hormone therapy affects cardiovascular risk in transgender persons has not been unraveled yet. In an attempt to do so, the effect of GAHT on different cardiometabolic markers has been studied. While results from different studies are inconsistent, testosterone in trans men seems to elevate lipid levels [3, 6] and blood pressure [2], and decrease insulin resistance [7, 8]. Diversely, estrogens (combined with antiandrogens) in trans women seem to have either a positive or no effect on the lipid spectrum [3, 6], and increase insulin resistance [7, 8]. Alterations in blood pressure in trans women most probably depend on factors as administration type, duration of hormone treatment and measurement method, as both increases as decreases are reported [2, 3]. In conclusion, current evidence is not able to define the mechanism by which cardiovascular risk is increased in transgender persons.

Two key players in the development of cardiovascular disease are the processes of atherosclerosis and hemostasis. Endogenous estrogens have beneficial effects on these processes; they promote vasodilatation and endothelial cell-growth and decrease the development of atherosclerosis in cis women [9]. In contrast, oral, but not transdermal, exogenous estrogens increase thrombotic risk in postmenopausal women [10], which may be the result of first-pass hepatic metabolism. Endogenous testosterone has both protecting and deleterious effects on the vasculature [2], and administration of exogenous testosterone does not clearly affect the risk of thrombosis in cis men [11]. As research on this topic in transgender persons is scarce, we aimed to explore the effects of GAHT on inflammation and hemostasis. We selected a broad spectrum of markers associated with systemic, endothelial, or adipose tissue related inflammation. In the context of hemostasis, platelet activation and coagulation markers were investigated.

## Materials and methods

### Study design

This is a prospective observational study, which is part of the ENIGI (European Network for the Investigation of Gender Incongruence) study. The ENIGI-study is conducted in four

collaborating gender clinics in Amsterdam, Ghent, Oslo and Florence. It is registered at https://clinicaltrials.gov/ct2/show/NCT01072825 and the full study protocol is published elsewhere [12]. The overall study protocol was approved by the ethical review board of Ghent University Hospital, Belgium and local ethical review boards of the other participating centers. Participants of the ENIGI-study are 18 years or older, diagnosed with gender dysphoria according to the revised fourth or fifth edition of the Diagnostic and Statistical Manual of Mental Disorders and receive routine clinical transgender care. Exclusion criteria are previous or current use of hormone treatment. Written informed consent is obtained.

In this study, data from participants included in Amsterdam between June 2012 and July 2019 was analyzed. All subjects were 50 years or younger, used transdermal hormone treatment and had available blood samples at baseline and after three and / or twelve months. Trans women received estradiol patches (100 mcg/24 hours, twice weekly) combined with CPA (50 mg daily) and trans men received testosterone gel (50 mg daily). If necessary, the dosage of the hormone treatment was adjusted to achieve adequate estradiol or testosterone concentrations as suggested by applying guidelines [1]. Exclusion criteria were use of anti-inflammatory medication or medication that affects hemostasis (e.g. platelet inhibitors, anticoagulation, SSRI's, etc.). Trans men were not allowed to use hormonal medication to suppress medication. None of the participants underwent gender-affirming genital surgery before or during the study-period. Follow-up duration was one year.

### Data collection

Venous blood samples were taken and Body Mass Index (BMI) and blood pressure were measured before the start of GAHT (baseline) and after 3 and 12 months of treatment. Estradiol (pmol/L) and testosterone (nmol/L) concentrations were measured at the Laboratory for Endocrinology of the Amsterdam University Medical Centers.

### Outcome measures

Our primary outcome measures consisted of several inflammatory markers. We selected a broad spectrum of markers in order to investigate different influences on inflammation, hemostasis and adipose tissue, the latter already known to be altered by hormone therapy in transgender persons [7, 13]. Of the examined systemic inflammatory markers, high-sensitivity C reactive protein (hs-CRP) is directly associated with cardiovascular risk [14] while cytokines may play an intermediate role in the acute phase response ($\alpha$-1-antitrypsin, tumor necrosis factor alpha (TNF-$\alpha$), interferon gamma (IFN-$\gamma$), interleukine (IL)-1b, IL-4, IL-5, IL-6, IL-8, IL-10, IL-12p70, IL-22) and vascular adhesion molecule 1 (VCAM-1) is expressed during endothelial activation. Adipose tissue specific marker leptin is associated with obesity and is a predictor of myocardial infarction [15, 16], while adiponectin has anti-inflammatory effects, like increasing insulin sensitivity [16, 17]. Of the examined coagulation markers, fibrinogen is associated with cardiovascular risk [18], and plasminogen activator inhibitor-1 (PAI-1) is altered by hormone therapy in postmenopausal cis women [19]. Platelet activation markers consisted of platelet-specific proteins platelet factor 4 (PF-4), $\beta$-thromboglobulin and p-selectin, which are released by platelets upon activation [20].

Secondary outcome measures were estradiol and testosterone levels, BMI and blood pressure. BMI and blood pressure are influenced by GAHT [21], and are associated with inflammation [22, 23]. They were included to rule out that observed changes in concentrations of inflammatory markers were actually explained by changes in BMI or blood pressure.

## Biomarker assays

The above-mentioned markers were measured at the Netherlands Organization for Applied Scientific Research (Leiden, The Netherlands), using serum (for PF-4 and β-thromboglobulin) and EDTA (for all other markers) plasma samples and miniaturized biomarker assays. The samples from transgender men and women were randomized and evaluated on the same plates. To minimize analytical variability, all measurements of a particular biomarker were carried out on the same day, and longitudinal samples of a subject (0, 3 and 12 months) were analyzed on the same plate (96 wells). More specifically, cytokine concentrations were determined with a CorPlex™ Cytokine Panel on an SP-X™ imaging system (Quanterix, Billerica, MA, USA). All other markers were quantified by enzyme-linked immunosorbent assay using the following antibody sets: hs-CRP (DY1707); VCAM-1 (DY809); adiponectin (DY1065); PF4 (DY795); β-thromboglobulin (DY393); p-selectin (DY137); PAI-1 (DY9387); all R&D Systems (Abingdon, UK); α-1-antitrypsin (NBP2-60541) and fibrinogen (NBP2-60465) from Novus Biologicals (Wiesbaden, Germany); leptin (10-1199-01) from Mercodia (Uppsala, Sweden). For each biomarker, the linear range and optimal dilution factor was optimized prior to the measurements using commercially available reference plasma from female (n = 20) and male blood donors (n = 20) (TCS Bioscience Ltd, Buckingham, UK). The linear range, limit of quantification and dilution factor of the inflammatory markers are provided in S1 Table.

## Statistical analysis

Statistical analyses were performed using STATA®, version 15.1. Baseline characteristics and hormone concentrations are presented as median with interquartile range (IQR). Concentrations of inflammatory markers were log transformed for analysis and back-transformed for presentation. Normality of the residuals was verified via visual inspection of the histogram. Linear mixed models [24] with a random intercept for each subject were used, with the inflammatory marker as the dependent variable and duration of treatment (0, 3 and 12 months) as a categorical covariate. We adjusted for potential confounders (change in BMI or blood pressure) and stratification factors (age, hormone level during treatment, smoking status). The obtained regression coefficient was back-transformed to the ratio and converted to percentage change. No adjustment for multiple comparisons was performed.

## Results

In this study, 48 trans women and 47 trans men were included. The study flowchart is shown in Fig 1. Table 1 shows the baseline characteristics of the study population. Hormone concentrations, body mass index and blood pressure at baseline and after 3 and 12 months of hormone therapy are shown in Table 2. Absolute values of the inflammatory markers are shown in Tables 3 and 4; percentage change is shown in Fig 2.

Below, Adjusting the analyses for change in BMI or blood pressure did not affect the results. Also, changes in markers were not different for different age ranges (in trans women only; in trans men no stratification was performed because of the small age range). Lastly, higher hormone concentrations during treatment or different smoking status at baseline did not affect the results either. Therefore, non-adjusted results are reported. We only describe the percentage change after 12 months of hormone therapy, unless the direction of the 3- and 12-month effect was different.

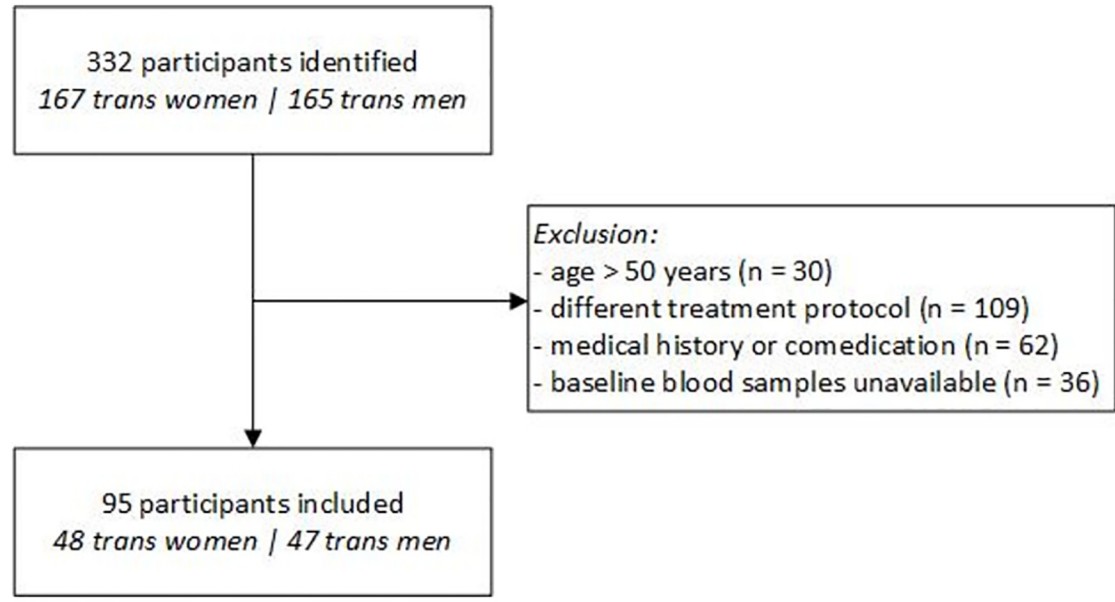

**Fig 1. Study flowchart.**

### Trans women

**Inflammation.** Systemic inflammatory markers IL-1b, IL4, IL5, IL12p70 and IFN-γ were undetectable in the majority of samples. Estimates of concentration changes of these markers are not considered reliable and are therefore not reported.

After 12 months of transdermal estradiol plus CPA, systemic and endothelial inflammatory marker concentrations decreased (hs-CRP -66%, (95% CI -76; -53), IL-6–28%, (95% CI -7; -44), IL-8–15%, (95% CI -2; -17), IL-22–34%, (95% CI -7; -44), VCAM-1–12%, (95% CI -16;

**Table 1. Baseline characteristics of the study population.**

|  | Trans women | Trans men |
|---|---|---|
|  | (n = 48) | (n = 47) |
| Age (years) | 30 (24–39) | 23 (20–26) |
| BMI (kg/m^2) | 23 (21–26) | 23 (21–30) |
| Ever smoked (%) | 16 (33%) | 20 (43%) |
| Blood pressure (mmHg) |  |  |
| Systolic | 129 (121–137) | 122 (115–128) |
| Diastolic | 80 (76–87) | 77 (72–82) |
| MAP | 96 (91–101) | 92 (87–97) |
| Glucose (mmol/L) | 5.5 (5.1–5.7) | 5.1 (4.9–5.4) |
| Total cholesterol (mmol/L) | 4.3 (3.7–5.1) | 4.3 (3.9–4.9) |
| Triglycerides (mmol/L) | 0.9 (0.7–1.3) | 0.7 (0.5–1.0) |

Data are presented as median (IQR) for continuous data, and number and % for categorical data. BMI, body mass index. MAP, mean arterial pressure. Number of participants with an underlying disease: diabetes mellitus (1), human immunodeficiency virus (2), pulmonal hypertension (1), ulcerative colitis (1).

**Table 2. Hormone concentrations, body mass index and blood pressure at baseline and after 3 and 12 months of hormone therapy.**

| | Trans women | | | Trans men | | |
|---|---|---|---|---|---|---|
| | Visit (months) | | | | | |
| | 0 | 3 | 12 | 0 | 3 | 12 |
| Estradiol (pmol/L) | 101 (79–126) | 246 (120–342) | 258 (178–496) | 273 (168–442) | 184 (147–250) | 158 (116–204) |
| Testosterone (nmol/L) | 18 (14–23) | 0.6 (0.5–0.8) | 0.7 (0.5–0.8) | 1.3 (1.0–1.7) | 27 (14–42) | 20 (15–30) |
| BMI (kg/m^2) | 23 (21–26) | 23 (21–26) | 24 (22–27) | 23 (21–30) | 23 (22–29) | 23 (22–28) |
| BP (mmHg) | | | | | | |
| Systolic | 129 (123–138) | 123 (117–134) | 127 (117–135) | 122 (115–126) | 123 (117–130) | 122 (116–129) |
| Diastolic | 80 (73–88) | 78 (73–83) | 78 (73–84) | 77 (72–81) | 75 (70–82) | 76 (72–82) |
| MAP | 96 (91–101) | 93 (88–98) | 94 (89–100) | 92 (87–97) | 90 (86–98) | 92 (87–96) |

Data are presented as median (IQR). BP, blood pressure. BMI, body mass index. MAP, mean arterial pressure.

-8)). Concentrations of both adipokines increased (leptin +202%, (95% CI 139; 282), adiponectin +7% (95% CI -2; 16)). For percentage change after 3 and 12 months, see Fig 2.

**Hemostasis.** Concentrations of platelet activation markers increased after 12 months of treatment (PF-4 +17%, (95% CI 4; 32), β-thromboglobulin +13% (95% CI 2; 24)). Coagulation

**Table 3. Inflammation and hemostasis markers at 0, 3 and 12 months in trans women (absolute values).**

| | Visit (months) | | | | |
|---|---|---|---|---|---|
| | 0 | 3 | | 12 | |
| **Inflammation** | | | | | |
| *Systemic inflammatory markers* | | | | | |
| hs-CRP (μg/ml) | 0.8 (0.6, 1.1) | 0.5 (0.3, 1.0) | p = 0.04 | 0.3 (0.2, 0.4) | p < 0.001 |
| α-1-antitrypsin (μg/ml) | 186 (173, 199) | 198 (177, 222) | p = 0.62 | 178 (165, 192) | p = 0.24 |
| TNF-α (pg/ml) | 2.9 (2.6, 3.3) | 2.5 (2.2, 2.9) | p = 0.06 | 2.6 (2.3, 2.9) | p = 0.02 |
| IL-6 (pg/ml) | 1.1 (0.9, 1.4) | 1.0 (0.7, 1.4) | p = 0.48 | 0.8 (0.6, 1.0) | p = 0.01 |
| IL-8 (pg/ml) | 6.3 (5.6, 7.0) | 5.9 (5.1, 6.9) | p = 0.52 | 5.3 (4.6, 6.1) | p = 0.02 |
| IL-10 (pg/ml) | 0.6 (0.5, 0.8) | 0.5 (0.4, 0.6) | p = 0.33 | 0.5 (0.4, 0.6) | p = 0.06 |
| IL-22 (pg/ml) | 0.8 (0.6, 0.9) | 0.5 (0.4, 0.7) | p = 0.33 | 0.5 (0.4, 0.7) | p = 0.06 |
| *Endothelial inflammatory markers* | | | | | |
| VCAM-1 (ng/ml) | 382 (358, 407) | 328 (299, 360) | p < 0.001 | 337 (315, 360) | p < 0.001 |
| *Adipose tissue markers* | | | | | |
| Leptin (ng/ml) | 3.5 (2.4, 5.1) | 7.6 (5.1, 11.4) | p < 0.001 | 9.8 (7.5, 12.8) | p < 0.001 |
| Adiponectin (μg/ml) | 2.3 (2.1, 2.6) | 2.6 (2.1, 3.2) | p = 0.015 | 2.5 (2.1, 2.8) | p = 0.13 |
| **Hemostasis** | | | | | |
| *Platelet activation markers* | | | | | |
| Platelet factor 4 (PF-4) (μg/ml) | 2.0 (1.8, 2.3) | 1.9 (1.4, 2.4) | p = 0.28 | 2.4 (2.2, 2.6) | p = 0.01 |
| β-thromboglobulin (μg/ml) | 5.6 (5.1, 6.1) | 5.3 (4.2, 6.5) | p = 0.35 | 6.4 (5.8, 6.9) | p = 0.02 |
| P-selectin (ng/ml) | 36 (32, 40) | 39 (35, 43) | p = 0.29 | 34 (30, 38) | p = 0.37 |
| *Markers of coagulation* | | | | | |
| Fibrinogen (mg/ml) | 3.7 (3.3, 4.1) | 4.3 (3.5, 5.2) | p = 0.04 | 3.8 (3.4, 4.1) | p = 0.78 |
| PAI-1 (ng/ml) | 23 (20, 26) | 24 (20, 30) | p = 0.46 | 21 (18, 25) | p = 0.45 |

Predicted average marker levels (geometric mean) with 95% confidence intervals and p-values, not adjusted for multiple comparisons. Measures were log-transformed for analysis and back-transformed for presentation.

**Table 4. Inflammation and hemostasis markers at 0, 3 and 12 months in trans men (absolute values).**

| | Visit (months) | | | | | |
|---|---|---|---|---|---|---|
| | **0** | **3** | | **12** | | |
| **Inflammation** | | | | | | |
| *Systemic inflammatory markers* | | | | | | |
| hs-CRP (µg/ml) | 0.3 (0.2, 0.6) | 0.5 (0.3, 0.8) | p = 0.23 | 0.5 (0.3, 0.9) | p = 0.005 | |
| α-1-antitrypsin (µg/ml) | 176 (165, 188) | 166 (150, 184) | p = 0.10 | 166 (150, 184) | p = 0.10 | |
| TNF-α (pg/ml) | 2.7 (2.4, 3.0) | 2.8 (2.5, 3.1) | p = 0.82 | 2.8 (2.5, 3.1) | p = 0.85 | |
| IL-6 (pg/ml) | 1.1 (0.9, 1.4) | 1.1 (0.8, 1.6) | p = 0.76 | 1.0 (0.8, 1.4) | p = 0.53 | |
| IL-8 (pg/ml) | 5.6 (4.8, 6.4) | 6.1 (5.4, 6.9) | p = 0.15 | 6.7 (5.7, 7.8) | p = 0.10 | |
| IL-10 (pg/ml) | 0.5 (0.4, 0.5) | 0.5 (0.4, 0.6) | p = 0.64 | 0.5 (0.4, 0.6) | p = 0.40 | |
| IL-22 (pg/ml) | 0.8 (0.6, 1.0) | 0.7 (0.5, 1.0) | p = 0.64 | 0.8 (0.6, 1.1) | p = 0.91 | |
| *Endothelial inflammatory markers* | | | | | | |
| VCAM-1 (ng/ml) | 386 (364, 410) | 407 (371, 446) | p = 0.13 | 390 (361, 421) | p = 0.37 | |
| *Adipose tissue markers* | | | | | | |
| Leptin (ng/ml) | 14.0 (10.8, 18.0) | 9.9 (7.1, 13.7) | p < 0.001 | 6.6 (4.9, 9.1) | p < 0.001 | |
| Adiponectin (µg/ml) | 3.1 (2.6, 3.6) | 2.4 (2.1, 2.9) | p < 0.001 | 2.5 (2.1, 2.9) | p < 0.001 | |
| **Hemostasis** | | | | | | |
| *Platelet activation markers* | | | | | | |
| Platelet factor 4 (PF-4) (µg/ml) | 2.1 (1.9, 2.3) | 1.7 (1.3, 2.2) | p = 0.04 | 2.1 (1.8, 2.4) | p = 0.85 | |
| β-thromboglobulin (µg/ml) | 5.4 (5.1, 5.9) | 4.4 (3.4, 5.8) | p = 0.03 | 5.5 (5.0, 6.1) | p = 0.99 | |
| P-selectin (ng/ml) | 33 (30, 37) | 36 (32, 40) | p = 0.56 | 34 (30, 38) | p = 0.72 | |
| *Markers of coagulation* | | | | | | |
| Fibrinogen (mg/ml) | 3.5 (3.2, 3.9) | 3.0 (2.3, 3.9) | p = 0.10 | 3.2 (2.8, 3.6) | p = 0.36 | |
| PAI-1 (ng/ml) | 23 (20, 25) | 26 (21, 32) | p = 0.40 | 21 (17, 25) | p = 0.35 | |

Predicted average marker levels (geometric mean) with 95% confidence intervals and p-values, not adjusted for multiple comparisons. Measures were log-transformed for analysis and back-transformed for presentation.

marker fibrinogen transiently increased after 3 months (+15%, (95% CI 1; 32)) and normalized after 12 months (+1%, (95% CI -8; 12)); PAI-1 was not altered. For percentage change after 3 and 12 months, see Fig 2.

### Trans men

**Inflammation.** After 12 months of testosterone treatment, concentrations of systemic inflammatory markers did not change, except for hs-CRP (+71%, (95% CI 19; 145)). Endothelial marker VCAM-1 was not clearly affected. Adipokine concentrations decreased (leptin -49% (95% CI -59; -37), adiponectin -20% (95% CI -27; -14)). For percentage change after 3 and 12 months, see Fig 2.

**Hemostasis.** Concentrations of platelet activation markers and coagulation markers did not clearly change after 12 months of treatment (PF-4 +0%, (95% CI -18; 23), β-thromboglobulin +2% (95% CI -16; 23), fibrinogen -8% (95% CI -23; 9)). For percentage change after 3 and 12 months, see Fig 2.

## Discussion

In this study, we assessed the association between hormone treatment and markers of inflammation and hemostasis in transgender persons. We observed that hormone treatment in trans

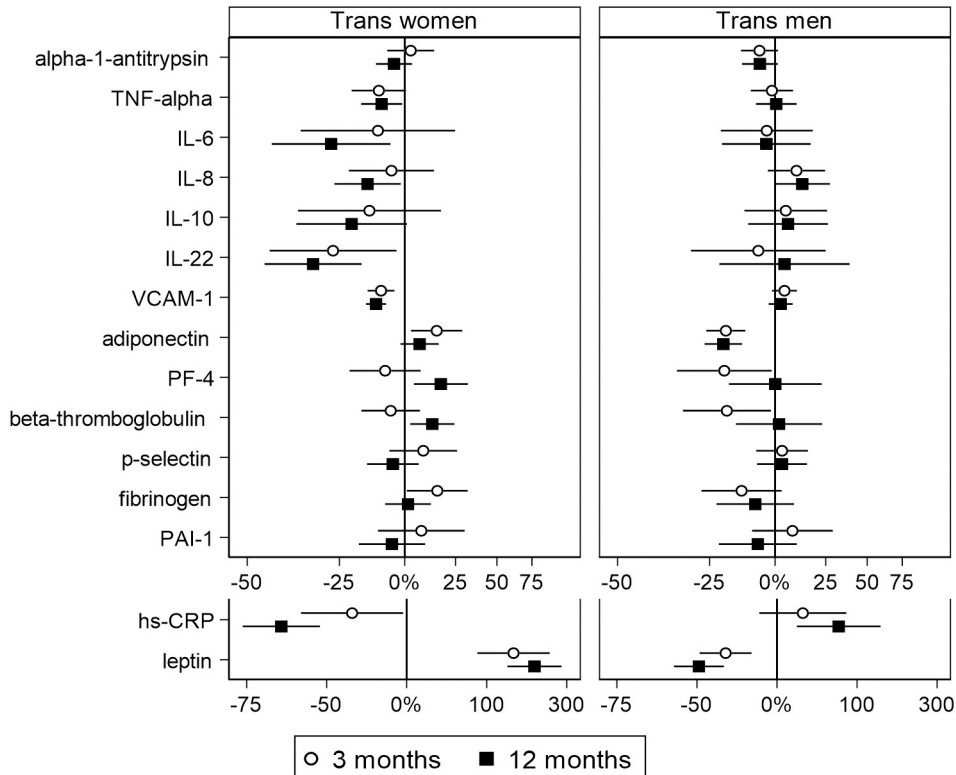

**Fig 2. %-change after 3 and 12 months of hormone therapy in trans women and trans men.** Values are presented as geometric mean with 95% confidence interval. Measures were log-transformed for analysis and back-transformed for presentation. Scales are logarithmic. For hs-CRP and leptin, a larger scale is used.

women was associated with a decrease in inflammatory marker concentrations and an increase in hemostasis marker concentrations. In contrast, in trans men, hormone treatment was associated with no change in inflammatory and hemostasis marker concentrations, except for an increase in hs-CRP.

## Trans women

**Inflammation.** Transdermal estradiol combined with CPA was associated with a decrease in concentrations of systemic and endothelial inflammatory markers in trans women. This is not in line with previous studies in postmenopausal and trans women, which have found that transdermal estradiol was not associated with a change in inflammation markers and oral estrogens were associated with an increase in inflammation marker concentrations [25–27]. However, both the postmenopausal and trans women included in these studies, were older than our participants. The potential beneficial effect of hormone therapy on inflammation decreases with age, as estradiol may not inhibit the development of atherosclerosis once this is present [28]. Further, the increase in concentrations of inflammatory markers associated with oral but not with transdermal estrogens implies that the hepatic first pass effect may be responsible.

In contrast with the other inflammatory markers, hormone treatment was associated with increased adipokine concentrations. These results are in line with previous studies in transgender persons using hormone treatment, which showed that concentrations of adipokines changed towards reference values of the experienced gender [7, 13]. The increased leptin

concentration may, at least partly, be explained by a change in fat distribution from visceral depots to the subcutaneous depot [29]. Subsequently, the change in adipokine concentrations can be interpreted as a consequence of metabolic adjustments in energy balance and body fat distribution.

**Hemostasis.** In trans women, transdermal estradiol plus CPA was associated with an increase in levels of platelet activation markers. This is in line with a previous study in cis women, which has shown that platelets express estrogen receptors and platelet activation differs per phase of the menstrual cycle [30]. However, studies in postmenopausal women are inconclusive, with some studies reporting an increase and other reporting a decrease in platelet activity after hormone treatment [31, 32].

The concentration of the coagulation marker fibrinogen transiently increased after 3 months of transdermal estradiol combined with CPA. Similarly, a recent study on the effect of GAHT on coagulation parameters observed more procoagulant profiles in trans women [33]. Our observed increase in fibrinogen concentration after 3 months of hormone therapy may indicate temporarily increased pro-coagulant activity. The transient increase in fibrinogen concentration could be caused by CPA. Oral contraceptives containing CPA are associated with a higher thrombotic risk than contraceptives containing levonorgestrel [34]. Also, the dosage of CPA that has antiandrogenic effects is higher than the dosage used in contraception (50 vs 2 mg), which may even be more prothrombotic. While we know that trans women using hormone treatment have an increased thrombotic risk, we do not know if those using CPA have a higher thrombotic risk than those using other (for example GnRH-analogues or spironolactone) or no antiandrogens.

### Trans men

**Inflammation.** In trans men, testosterone was associated with an increase in hs-CRP concentration, but not with a change in other systemic inflammatory markers. This is in contrast with previous studies in hypogonadal cis men, which found no effect or even a decrease in levels of inflammation markers [35, 36]. This difference may be explained by the effect of testosterone administration on estrogen concentrations. In hypogonadal cis men, testosterone is converted into estradiol by aromatase, increasing the concentration of estradiol. However, trans men start with a high concentration of estradiol, which is decreased by testosterone Since our results in trans women suggest that estrogens are associated with a decrease in inflammatory markers the increase of hs-CRP in trans men may be associated with the decline in estrogen concentrations. This however does not explain why we observed no change in concentrations of the other inflammatory markers.

Further, we found that testosterone decreased adipokine levels towards reference values for men, which is in line with our results in trans women and the results of previous studies in transgender persons [7, 13]. This is probably partially explained by a change in fat distribution from subcutaneous towards visceral fat depots [29].

**Hemostasis.** While estradiol plus CPA increased hemostatic marker concentrations in trans women, testosterone did not affect these markers in trans men. After 12 months of hormone treatment, platelet activation marker concentrations did not change. Unfortunately, previous studies on this topic only include animal and ex vivo studies. Some of these studies have suggested that testosterone induces platelet aggregation by influencing platelet receptor expression [37, 38], while another indicated that testosterone induces platelet inhibition [39]. Therefore, the effect of testosterone on platelet activity needs to be further examined.

Administration of testosterone also did not affect coagulation markers in trans men. This is in line with a recent study on coagulation parameters in trans men, which observed no

apparent changes [33]. Also, there is absence of evidence for an increased risk of venous thromboembolic disease in trans men [2–5]. Similarly, in hypogonadal cis men, testosterone replacement therapy does not increase concentrations of coagulation markers [40], nor the occurrence of venous thromboembolism [11]. While our results indicate that the hemostatic system may play a role in the association between hormone treatment and cardiovascular risk in trans women, this is less apparent in trans men.

## Strengths and limitations

As far as we know, this is the first study exploring the effect of gender-affirming hormone therapy on inflammatory and hemostasis markers in both trans women and trans men. The explorative aim of our study required investigating multiple markers and multiple testing. Therefore, our results are hypothesis-generating, and not hypothesis-confirming. A strength of our study is the relatively large homogenous population and the specific hormone regimes and administration types. As it is unethical to include a control group who are withheld from desired hormone therapy, we used a prospective design in which each participant serves as their own control. Our study had some limitations as well. First, our participants were young and healthy and follow-up duration was relatively short, while the occurrence of atherosclerosis and cardiovascular disease increases with age. Second, we were unable to differentiate between the effect of transdermal estradiol and the effect of CPA in trans women. Third, to avoid the first-pass effect of the liver, we investigated the effect of only transdermal estradiol, while the occurrence of thrombosis is especially associated with the use of oral estrogens. Future, larger studies in older transgender persons, with a longer follow-up duration, different hormone regimes and administration types, preferably in comparison to cisgender controls, are necessary to expand our knowledge on this subject. While especially relevant for the trans population, gaining insight into the mechanism by which sex hormones affect cardiovascular risk may help to understand sex differences in cardiovascular disease in the cis population as well.

## Supporting information

**S1 Table. Linear range, limit of quantification and dilution factor of inflammatory markers.**
(DOCX)

**S2 Table. Number of analyzed blood samples for trans women and trans men at 0, 3 and 12 months.**
(DOCX)

**S1 Appendix. Minimal dataset.**
(DTA)

## Acknowledgments

We would like to express our gratitude towards all individuals who participated in the study.

## Author Contributions

**Conceptualization:** Moya H. Schutte, Robert Kleemann, Guy T'Sjoen, Abel Thijs, Martin den Heijer.

**Data curation:** Jessica M. Snabel.

**Formal analysis:** Chantal M. Wiepjes, Jessica M. Snabel.

**Funding acquisition:** Robert Kleemann.

**Methodology:** Moya H. Schutte, Robert Kleemann, Chantal M. Wiepjes.

**Resources:** Robert Kleemann.

**Supervision:** Abel Thijs, Martin den Heijer.

**Writing – original draft:** Moya H. Schutte.

**Writing – review & editing:** Robert Kleemann, Nienke M. Nota, Guy T'Sjoen, Abel Thijs, Martin den Heijer.

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
