## [Decision Letter · Decision Letter 0]

15 Oct 2021

PONE-D-21-30629The effect of transdermal gender-affirming hormone therapy on markers of inflammation and hemostasisPLOS ONE

Dear Dr. Schutte,

Thank you for submitting your manuscript to PLOS ONE. After careful consideration, we feel that it has merit but does not fully meet PLOS ONE’s publication criteria as it currently stands. Therefore, we invite you to submit a revised version of the manuscript that addresses the points raised during the review process.

The research has potential interest for the journal, but the manuscript has serious deficiencies and mistakes that must be attended, I suggest attending the observations of the reviewer 1, mainly about statistical analysis, and consider the suggests of reviewer 2. A second review of the manuscript and point-to point response to reviewers are necessary. 

We look forward to receiving your revised manuscript.

Kind regards,

Martha Asuncion Sánchez-Rodríguez, PhD

Academic Editor

PLOS ONE

Journal Requirements:

"The biochemical analyses were supported by the TNO research programs Biomedical Health-PMC13 and Prevention 2.0. The funders had no role in study design, data collection and analysis, decision to publish, or preparation of the manuscript."

Additional Editor Comments:

The research has potential interest for the journal, but the manuscript has serious deficiencies and mistakes that must be attended. I suggest attending the observations of the reviewer 1, mainly about statistical analysis, and consider the suggests of reviewer 2. A second review of the manuscript and point-to point response to reviewers are necessary.

Reviewers' comments:

Reviewer's Responses to Questions

**Comments to the Author**

1. Is the manuscript technically sound, and do the data support the conclusions?

Reviewer #1: Partly

Reviewer #2: Yes

2. Has the statistical analysis been performed appropriately and rigorously? 

Reviewer #1: Yes

Reviewer #2: Yes

3. Have the authors made all data underlying the findings in their manuscript fully available?

Reviewer #1: Yes

Reviewer #2: Yes

4. Is the manuscript presented in an intelligible fashion and written in standard English?

Reviewer #1: No

Reviewer #2: Yes

5. Review Comments to the Author

Reviewer #1: Schutte et al looked at the effect of transdermal GAHT on markers of inflammation,

The study is nicely done but the authors need to recognize the limitation of their study. The sample size is very small and no control subjects but 15 markers have been tested. This is a big limitation for the study, the authors should mention that this is an exploratory study and no way is a confirmatory study. The english used in the text is also hard to follow and needs rewriting. The abbreviation needs to be spelled out first time used in the body of the manuscript, for example "PAI-1" never been spelled out.

There are my comments about the statistical analysis:

1. Authors report that “Concentrations of inflammatory markers were log transformed for analysis and back transformed for presentation”. While this is probably correct, authors should verify that the residual errors from the mixed linear model have a normal distribution on the log scale by computing a normal goodness of fit statistic (for example Shapio-Wilk test) and report that normality was verified. If the authors are reporting the antilog of the log scale means, this is called the geometric mean and should be labelled as such.

2. Data (IL-1b, IL-4,IL12p70) failing to have a normal distribution (“heavily right skewed”-line 234) is not a good reason for omitting results (“they were not further analyzed”). Non parametric methods can be used. Medians and interquartile ranges can be reported. They can probably ignore these markers if they think they are not important markers otherwise their reason for not analyzing is not acceptable.

3. Authors should report the sample size at each time if it is not constant (48 trans females or 47 in trans males) across time.

4. There is possible selection bias if those with no baseline data were excluded. Authors should provide the flow diagram showing how many were excluded from the initial pool and why they were excluded.

5. While the authors did not report BMI or blood pressure change adjusted results, it is not clear why this should even be considered. It would seem that change in BMI and/or blood pressure are not confounders but are another result of the hormone intervention. This needs to be clarified (especially for BP)

6. The authors looked at 15 outcomes (Table3). They should report nominal p values for the change from time 0 and state whether they are adjusting for multiple outcomes and multiple comparisons. The authors are commended for reporting confidence bounds but should clarify that the 95% confidence level is not adjusted for multiple outcomes.

Obviously, a stronger study would compare these changes to changes in a cisgender control group but it is acknowledged that this may not be feasible and is beyond the scope of the current study.

7-It would also have been interesting if the change in the outcome had been correlated with the change in hormone but this may be beyond the scope of the current analysis.

Reviewer #2: This is an interesting report about the effect of hormone therapy on markers of inflammation and hemostasis in transgender persons. The findings were clear-cut; however, the manuscript has some problems.

The keyword cardiovascular events should be revised

Abstract

The result about the increment in platelet factor 4 (PF-4) in trans women is not mentioned

Introduction

It is not clear why the authors measure leptin and adiponectin, they should explain in a little more detail why they chose these adipokines as adipose tissue markers

Materials and methods

The sentence in line 154 “Using EDTA or serum plasma samples” should be revised

Results

Authors did not mention whether the dosage of the hormone therapy was adapted in some participants to reach adequate estradiol or testosterone concentrations as suggested by guidelines

The authors omit important characteristics of the study population as: blood pressure, weight, BMI, glucose levels, and lipid profile at baseline and after 3 and 12 months of hormone therapy

There are no data concerning the liver and renal function for the safety evaluation

It is not specified if some participants had an underlying disease

In table 3 the authors should provide the p-values of all the comparisons

Figure 1 displays a reduction in IL-6, IL-8 and IL-22 in trans women at 12 months; these results are not mentioned in results section and are not discussed

6. PLOS authors have the option to publish the peer review history of their article (what does this mean?). If published, this will include your full peer review and any attached files.

Reviewer #1: **Yes: **Leila Hashemi MD MSCR FACP

Reviewer #2: **Yes: **Renata Saucedo

---

## [Author Response · Author response to Decision Letter 0]

19 Nov 2021

We would like to thank the editor and the reviewers for the helpful suggestions, and for providing us the opportunity to resubmit our manuscript entitled "The effect of transdermal gender-affirming hormone therapy on markers of inflammation and hemostasis". We were delighted to receive such thoughtful feedback, and we feel that our manuscript has improved substantially. We have attached a point-by-point reply to the comments, and our revised manuscript.

---

## [Decision Letter · Decision Letter 1]

1 Dec 2021

The effect of transdermal gender-affirming hormone therapy on markers of inflammation and hemostasis

PONE-D-21-30629R1

Dear Dr. Schutte,

We’re pleased to inform you that your manuscript has been judged scientifically suitable for publication and will be formally accepted for publication once it meets all outstanding technical requirements.

Kind regards,

Martha Asuncion Sánchez-Rodríguez, PhD

Academic Editor

PLOS ONE

Additional Editor Comments (optional):

Thanks to attend the reviewers’ comments, the manuscript substantially improves.

Reviewers' comments:

Reviewer's Responses to Questions

**Comments to the Author**

1. If the authors have adequately addressed your comments raised in a previous round of review and you feel that this manuscript is now acceptable for publication, you may indicate that here to bypass the “Comments to the Author” section, enter your conflict of interest statement in the “Confidential to Editor” section, and submit your "Accept" recommendation.

Reviewer #2: All comments have been addressed

2. Is the manuscript technically sound, and do the data support the conclusions?

Reviewer #2: Yes

3. Has the statistical analysis been performed appropriately and rigorously? 

Reviewer #2: Yes

4. Have the authors made all data underlying the findings in their manuscript fully available?

Reviewer #2: Yes

5. Is the manuscript presented in an intelligible fashion and written in standard English?

Reviewer #2: Yes

6. Review Comments to the Author

Reviewer #2: I am satisfied with the authors responses and with the modifications made to the original manuscript.

7. PLOS authors have the option to publish the peer review history of their article (what does this mean?). If published, this will include your full peer review and any attached files.

Reviewer #2: **Yes: **Renata Saucedo

---

## [Editor Report · Acceptance letter]

2 Mar 2022

PONE-D-21-30629R1 

The effect of transdermal gender-affirming hormone therapy on markers of inflammation and hemostasis 

Dear Dr. Schutte:

I'm pleased to inform you that your manuscript has been deemed suitable for publication in PLOS ONE. Congratulations! Your manuscript is now with our production department. 

Kind regards, 

on behalf of

Dr. Martha Asuncion Sánchez-Rodríguez 

Academic Editor

PLOS ONE